# Counter-Therapeutic Strategies for Resistance of FLT3 Inhibitors in Acute Myeloid Leukemia

**DOI:** 10.3390/cells14191526

**Published:** 2025-09-30

**Authors:** Moo-Kon Song

**Affiliations:** Division of Hematology-Oncology, Department of Internal Medicine, Haeundae Bumin Hospital, Busan 48094, Republic of Korea; mksong9676@hanmail.net; Tel.: +82-51-1677-0082

**Keywords:** acute myeloid leukemia, FMS-like tyrosine kinase 3, therapeutic resistance, tyrosine kinase inhibitors

## Abstract

FMS-like tyrosine kinase 3 (FLT3) mutations in acute myeloid leukemia (AML) are associated with an increased risk of relapse and a poor prognosis. Several FLT3 inhibitors that have been developed demonstrated efficacy against the FLT3 tyrosine kinase domain and/or internal tandem duplication mutations. Nevertheless, remission rates for these agents remain in the range of 30~40% of patients, attributed to both primary and secondary mechanisms of resistance, with relapse rates varying from ~30 to 50%. The mechanisms underlying resistance to FLT3 inhibitors have been characterized, offering valuable insights that can guide the development of clinical trials aimed at discovering novel FLT3 tyrosine kinase inhibitors (TKIs) that can overcome resistance. Additionally, elucidating resistance signaling pathways may facilitate the identification of other TKIs, rational combination therapies or multiple targeted TKIs to address alternative pathways, potentially helping overcome resistance in AML patients with refractory clones.

## 1. Introduction

FMS-like tyrosine kinase 3 (FLT3) is a receptor tyrosine kinase that contains distinct domains essential for its functional role in acute myeloid leukemia (AML) [1]. FLT3 functions in cellular signaling processes through activation by its ligand, FLT3 ligand (FL). Upon FL binding, dimerization of the FLT3 extracellular domain induces activation of the intracellular kinase domains, initiating signaling pathways that regulate cell survival, proliferation and differentiation [2].

FLT3 mutations are generally classified as internal tandem duplications (ITDs), characterized by duplications within the JM domain of the FLT3 receptor tyrosine kinase, and by FLT3-tyrosine kinase domain (TKD) mutations, which are point mutations occurring within the tyrosine kinase domain [3]. Both FLT3-ITD and FLT3-TKD mutations result in constitutive activation of several downstream signaling cascades [4]. FLT3 phosphorylation initiates various intracellular signaling pathways that regulate cell survival, proliferation and differentiation [5]. Moreover, FLT3 mutations can activate downstream pathways such as PI3K-AKT, RAS/MAPK and STAT5, which contribute to unregulated cell proliferation and impaired differentiation. Consequently, the presence of specific FLT3 mutations has been correlated with a poor prognosis in AML and has influenced the development of several FLT3 inhibitors [4].

Type I inhibitors including gilteritinib, midostaurin, crenolanib and lestaurtinib permit receptor activation following their entry into the ATP-binding site, thereby exerting activity against both FLT3-ITD and TKD mutations. In contrast, type II inhibitors such as gizartinib and sorafenib interact with the receptor in its inactive state, thereby preventing FLT3 activation and demonstrating potential efficacy only against FLT3-ITD mutations [6].

However, only about one-third of AML patients respond to FLT3 inhibitors. Primary resistance mechanisms to FLT3 inhibitors involve FL activation, increased expression of fibroblast growth factor 2 (FGF2), upregulation of CYP3A4 and elevated CXC chemokine receptor type 4 (CXCR4) [7]. In contrast, secondary resistance mechanisms are linked to the activation of alternative signaling pathways that bypass the inhibited FLT3 axis after prolonged FLT3 inhibitor exposure, which maintains continued cell proliferation and survival despite FLT3 pathway suppression.

Therefore, strategies to overcome resistance mediated by secondary FLT3 mutations are required. This review provides a summary of FLT3 inhibitor resistance mechanisms and discusses various approaches to overcome such resistance in AML.

## 2. Wild-Type FLT3 Signaling Pathways

Wild-type FLT3 (WT-FLT3) primarily exists as a monomer in its inactive form. Upon binding to FL, the receptor can dimerize, which leads to receptor activation and downstream signaling events. Dimerization induced by FL is essential for FLT3 activation and subsequent cellular effects [8,9]. In WT-FLT3, the activation loop (A-loop) is found within the TKD and plays a pivotal role in controlling receptor function. The A-loop may contain tyrosine residues that become phosphorylated following FL engagement. Tyrosine phosphorylation is essential for FLT3 dimerization, activation and the initiation of downstream signaling pathways.

Subsequent to FL binding, the FLT3 receptor initiates a signaling cascade involving Src homology 2 (SH2) domain-containing proteins, such as growth factor receptor-bound protein 2 (GRB2) and the Shc protein family, which then activate downstream pathways including RAS/MAPK and PI3K/AKT, key mediators of cell proliferation, survival and differentiation [10]. ERK1/2 activation, another downstream process, is associated with increased cell growth and proliferation through phosphorylation-dependent activation of transcription factors [11]. Notably, the PI3K/AKT pathway facilitates mTOR phosphorylation, enhancing protein synthesis and regulating genes that control proliferation and survival. Furthermore, Akt can inhibit apoptosis by promoting the stability of the anti-apoptotic protein Mcl-1 and phosphorylating the pro-apoptotic protein BAD.

## 3. Signaling Pathways Associated with FLT3 Mutations

FLT3-ITD mutations may activate downstream pathways such as PI3K/AKT and RAS/MAPK, which in turn facilitate the development and proliferation of leukemic cells (Figure 1). Additionally, the SH2 domain of STAT5 can dock onto the tyrosine residues Y589 and Y591 in the JM domain exposed by ITD, thereby initiating the STAT5 signaling pathway [12]. Activated STAT5 promotes the transcription of various genes involved in leukemic cell proliferation and survival, including cyclin D1, C-MYC, p21 and PIM. PIM then induces the phosphorylation and activation of serine residues on Cdc25A, C-MYC and Notch-1, and it modulates the pro-apoptotic protein BAD, collectively contributing to enhanced cell proliferation and the inhibition of apoptosis [13]. Conversely, inhibition of PIM-1 results in reduced STAT5 activity. Combined inhibition of PIM-1 and FLT3-ITD may act synergistically to induce apoptosis.

The second most prevalent form of FLT3 mutation in AML involves point mutations within the TKD. Mutations occurring in the A-loop constitute the most frequent alterations in TKD, and they are mainly located at residues D835, I836 and Y842 [14]. The A-loop, when in its inactive state, prevents the binding of ATP and substrates to the kinase domain. Consequently, point mutations disrupt this inhibitory function of the A-loop, resulting in constitutive kinase activation and persistent signaling through the RAS/MAPK and PI3K/AKT pathways.

## 4. Primary Resistance

Primary resistance to FLT3 inhibitors is influenced by several molecular signaling pathways, the BM stromal microenvironment and distinct metabolic processes. Elevated FL levels sustain persistent activation of the FLT3/MAPK pathway, continuously promoting leukemic cell survival (Figure 1). Subsequent hyperactivation of the MAPK/AKT pathway following increased FL levels leads to phosphorylation of the anti-apoptotic protein BAD, which enhances the survival and proliferation of leukemic cells [15]. Furthermore, a preclinical study demonstrated that exogenous FL decreases the effectiveness of lestaurtinib, midostaurin and sorafenib in MOLM14 cells [16]. In addition, elevated FL levels have been linked to increased activation of STAT1, STAT3 and STAT5, thereby promoting resistance to FLT3 inhibitors. Other data have shown an association between high FL expression and increased STAT5 activation [17]. The STAT signaling pathways have also been implicated in the development of other mechanisms of resistance to FLT3 inhibitors.

FGF2 promotes activation of fibroblast growth factor receptor 1 (FGFR1) and downstream MAPK signaling pathways, resulting in sustained activation and resistance to FLT3 inhibitors (Figure 1) [18]. Moreover, cytochrome P450 3A4 (CYP3A4), which is highly expressed in the liver and BM, plays a critical role in the metabolism of FLT3 inhibitors. CYP3A4 contributes significantly to BM stromal cell-mediated resistance to FLT3 inhibitors such as sorafenib, quizartinib and gilteritinib in FLT3-ITD-positive AML. CYP3A4 expression in BM stromal cells is thought to be a key factor underlying resistance to FLT3 inhibitors including sorafenib, quizartinib and gilteritinib [18,19]. In contrast, knockdown of CYP3A4 by the CYP3A4 inhibitor clarithromycin markedly decreased resistance to FLT3 inhibitors and overcame the protective effects of BM stromal cells.

An increase in CXC motif chemokine ligand 12 (CXCL12)-abundant cells is observed in FLT3-ITD-positive AML. CXCR4, which is expressed on leukemic cells, has also been shown to mediate proliferative and anti-apoptotic signaling in these cells (Figure 1). The interaction between FLT3-ITD-positive AML cells and BM stroma via the CXCL12/CXCR4 axis has been demonstrated to enhance resistance to FLT3 inhibitors [20]. Accordingly, activation of the CXCL12/CXCR4 pathway may facilitate the rapid development of resistance to FLT3 inhibitors. Conversely, suppression of CXCL12 increases the susceptibility of AML cells to chemotherapy regimens containing FLT3 inhibitors.

## 5. Secondary On-Target Resistance

During secondary resistance to FLT3 inhibitors, on-target resistance emerges from molecular alterations within the FLT3-ITD allele. This resistance is characterized by reduced efficacy of FLT3 inhibitors due to acquired abnormalities within FLT3, including additional FLT3 mutations and functionally defective factors that do not involve FLT3 [21].

In FLT3-mutated AML patients receiving quizartinib, secondary mutations at D835 and Y842 residues, as well as mutations in the F691 “gatekeeper” residue, have been identified. Gatekeeper mutations such as F691L are capable of conferring resistance to all FLT3 inhibitors, including midostaurin and quizartinib, by preventing inhibitor binding [22]. Mutations within the tyrosine kinase domain, such as N676, can also impair the activity of midostaurin. FLT3-TKD point mutations that affect the gatekeeper residue F691, as well as alterations in the activation loop (A-loop) at D835 or Y842, are implicated in acquired resistance to FLT3 inhibitors such as quizartinib. The A-loop mutations have additionally been detected in patients treated with sorafenib and other FLT3 inhibitors. However, crenolanib has not been associated with the development of A-loop mutations. Therefore, crenolanib may be beneficial in cases where A-loop mutations have arisen from prior TKI therapy [23].

## 6. Secondary Off-Target Resistance

Off-target resistance is associated with abnormal signaling that mediates constitutive activation of oncogenic pathways independent of FLT3. Clonal evolution during FLT3 inhibitor therapy may lead to the emergence of RAS-driven oncogenic mutations in patients with FLT3-ITD mutations. Additionally, the loss of FLT3-ITD mutation in relapsed cases may be related to “off-target” resistance mechanisms [24].

Activated mutations within the RAS/MAPK pathway may influence disease progression in relapsed or refractory AML patients receiving FLT3 inhibitor-based therapy. The RAS/MAPK pathway can also manifest in progressive disease treated with FLT3 inhibitors [25].

Upregulation of the JAK/STAT pathway has also been implicated in resistance mechanisms to FLT3 inhibitors [26]. Resistance develops through increased phosphorylation of multiple factors within the downstream pathway of the FLT3 receptor. FLT3-TKD mutations may also induce activation of SHP1 and SHP2, which act as negative regulators of the JAK signaling pathway. Such mutations can promote the development of a more aggressive AML phenotype harboring TKD mutations.

AXL activation may cause FLT3 resistance by triggering the RAS/MAPK and PI3K/AKT pathways. AXL is stimulated by growth arrest-specific factor 6 (GAS6) [27]. GAS6 activation leads to AXL dimerization, autophosphorylation and stimulation of downstream signaling cascades such as PI3K/AKT, RAS/MAPK, STAT and NF-κB [28]. Therefore, upregulation and activation of AXL facilitate leukemic cell proliferation and confer resistance to FLT3 inhibitors including midostaurin and quizartinib. AXL-1 activation has also been observed in sorafenib-resistant cells carrying D835Y mutations. Moreover, research indicates that various cytokines and BM hypoxia enhance AXL-1 expression through STAT5 signaling, which is associated with resistance to quizartinib [29].

FLT3-ITD mutation also promotes STAT5 activation through a conformational change in FLT3. STAT5 may enhance PIM-2 gene expression, and PIM-2 helps sustain leukemic cell proliferation by opposing FLT3 inhibition. Previous findings identified PIM-2 as a putative resistance mechanism to sorafenib. Furthermore, co-occurrence of FLT3-ITD and TKD mutations results in hyperactivation of STAT5 signaling via suppressor of cytokine signaling 1 (SOCS1) and SCOS2 [30]. STAT5 activation subsequently enhances the upregulation of PI3K, RAS, B cell lymphoma 2 and DNA repair protein homolog 1, thereby promoting resistance to FLT3 inhibitors.

## 7. Strategies to Overcome Resistance to FLT3 Inhibitors

### 7.1. Novel FLT3 Inhibitors

Sitravatinib is a potent FLT3 inhibitor that demonstrates substantial efficacy against FLT3 mutations, even in the presence of cytokines, by providing more potent and sustained inhibition of phosphorylated ERK and AKT (Table 1 and Figure 1). Sitravatinib has also demonstrated activity in overcoming FLT3 inhibitor-associated resistance, such as the F691L mutation, by targeting downstream signaling molecules including STAT5, AKT and ERK. Additionally, a preclinical study indicated that sitravatinib could act synergistically with venetoclax, a BCL-2 inhibitor, by reducing MCL-1 expression and inhibiting the MAPK and AKT pathways [31,32].

Ningetinib is a novel FLT3 inhibitor that is capable of overcoming secondary drug resistance in leukemic cells harboring multiple FLT3 mutations (Table 1 and Figure 1) [33]. It is effective against various FLT3 mutations, including TKD mutations such as D835Y/V and Y842C, as well as F691L. This agent inhibits phosphorylation of FLT3 and its downstream signaling pathways, including AKT, ERK and STAT5.

FF-10101 is the first irreversible FLT3 inhibitor, exhibiting significant efficacy in overcoming resistance to FLT3 inhibitors due to FLT3 F691L and D835 mutations (Table 1 and Figure 1) [34]. It irreversibly inhibits FLT3 kinase by covalently binding to the C695 residue [35]. Hence, FF-10101 may be effective for mutations at the C695 binding site and the Y693C mutation.

**Table 1 cells-14-01526-t001:** Scheme of novel tyrosine kinase inhibitors for resistance of FLT3 inhibitors in AML.

Novel Inhibitor	Action Mechanism	Clinical Impact of Inhibitors	Agents	References
Novel FLT3 inhibitors	Secondary resistance mutations such as F691L, D835 and Y842C could activate downstream pathways such as AKT, ERK and STAT5	Novel FLT3 inhibitors were effective against secondary resistance mutations of FLT3, thereby inhibiting downstream pathways	Sitravatinib Ningetinib FF-10101	[32,33,34]
AXL inhibitors	AXL activates ERK/STAT5, increases AML cell survival and has bypassing effects of FLT3 inhibitions	AXL inhibitors bind to gatekeeper mutations such as F691L and overcome resistance	TP-0903 CTS2016	[36]
HDAC inhibitors	HDAC8 downregulates p53 and increases leukemic cell survival; in addition, it reduces DNA repair defects in FLT3-ITD mutations	HDAC inhibitors selectively inhibit DNA repair in FLT3-ITD mutated cells, which induces HSP90 acetylation, FLT3 ubiquitination and proteasomal degradation of FLT3, thereby inhibiting FLT3 downstream pathways	IHCH9033 Compound 22d Compound 25h	[37,38,39]
CDK4/6 inhibitors	CDKs—serine/threonine protein kinases—are crucial to regulate cell cycle; they activate after binding to cyclins and regulate DNA replication in leukemic cells	CDK4/6 inhibitors could inhibit DNA replication and induce leukemic cell arrest	Palbociclib Abemaciclib AMG925	[40,41]
Dual JAK2 and FLT3 inhibitors	JAK2 mutation activates STAT5, MAPK and PI3K regardless of FLT3 action and activates bypass pathway of FLT3, thus achieving resistance to FLT3 inhibitors	Dual JAK2/FLT3 inhibitor -inhibits STAT5, MAPK and PI3K -suppress A-loop mutations (D835, D839, Y842)	Pacritinib Momelotinib Pluripontin Compound 11r	[42]
PIM inhibitor	PIM-1 upregulates downstream pathways of FLT3-ITD and promotes cell growth and survival, thus reducing sensitivity of FLT3 inhibitors	PIM inhibitors reduce replication of RNA and reduce Mcl-1 transcripts	SGI-1776 AZD1208duak JAK2	[43]
CXCR4 inhibitor	CXCL12/CXCR4 axis influences cell migration, adhesion and survival in BM The axis activates downstream MAPK and ERK pathway	CXCR4 inhibitors could counteract activation of downstream pathway such as MAPK and ERK	LYS10924 GMI-1359	[44]
BET inhibitor	BET associated with pro-survival factors MYC and BCL2	BET inhibitors have inhibitory effect on MYC and BCL2 and could overcome mutations in F691L and D835	JQ1 PLX51107	[45]

### 7.2. AXL Inhibitors

AXL activation triggers the reactivation of downstream signaling pathways, such as ERK and STAT5, contributing to leukemic cell survival and proliferation by bypassing the effects of FLT3 inhibition (Figure 1) [36]. Consequently, AXL inhibitors may play a role in overcoming resistance to FLT3 inhibitors in AML. FLT3-ITD mutations can acquire resistance to FLT3 inhibitors, and AXL is known to be upregulated and activated in response to FLT3 inhibitors, thus providing an alternative survival pathway for leukemic cells (Table 1 and Figure 1). Combined targeting of AXL and FLT3, either through combination therapy or dual AXL/FLT3 inhibitors, has the potential to circumvent resistance and enhance treatment responses. 

TP-0903 is a novel AXL inhibitor that also inhibits FLT3 activity. This agent has demonstrated promising in vitro and in vivo activity in AML cases resistant to FLT3 inhibitors [46]. It was specifically designed to bind the gatekeeper leucine residue in AXL, thereby helping to overcome resistance related to the clinically relevant FLT3 gatekeeper F691 leucine substitution mutation.

CTS2016 is an oral small-molecule inhibitor that exhibits nanomolar enzymatic activity against both AXL and FLT3. Suppressing AXL phosphorylation with CTS2016 was found to overcome resistance to FLT3 inhibitors in AML [47]. When combined with venetoclax or azacitidine, it demonstrated potent inhibitory effects and strongly induced cell death in FLT3-mutated AML cells [47].

DAXL-88 and DAXL-88-MMAE are also AXL inhibitors developed to overcome FLT3 inhibitor resistance in leukemic cells. Preclinical data indicate that these agents, in combination with FLT3 inhibitors, exert synergistic effects in FLT3 inhibitor-resistant cell lines [48].

### 7.3. Histone Deacetylase Inhibitors

Histone deacetylase (HDAC) inhibitors have been shown to overcome resistance to FLT3 inhibitors in AML cells. Furthermore, when combined with FLT3 inhibitors, HDAC inhibitors may improve FLT3 inhibitor efficacy by targeting diverse mechanisms of resistance. FLT3 inhibition can lead to the upregulation of HDAC8 via activation of FOXO1 and FOXO3 transcription factors [37]. HDAC8 deacetylates and inactivates the tumor suppressor protein p53, whose normal function is to facilitate apoptosis in leukemic cells. By causing p53 inactivation, HDAC8 supports leukemic cell survival and persistence despite FLT3 inhibitor therapy [37]. In addition, HDAC inhibitors address other mechanisms of resistance, including deregulated signaling pathways or DNA repair abnormalities, which are not directly affected by FLT3 inhibitors. Therefore, the co-administration of these agents with FLT3 inhibitors results in a synergistic effect, exceeding the efficacy achieved by individual agents alone (Table 1 and Figure 1).

IHCH9033, a novel selective class I HDAC inhibitor, demonstrated greater anti-tumor efficacy in FLT3-ITD-mutated AML. This agent can selectively suppress DNA repair processes in FLT3-ITD-mutated AML cells, resulting in accumulation of DNA damage, cell cycle arrest and apoptosis [38]. In addition, IHCH9033 can induce acetylation of HSP90, leading to the inactivation of the chaperone activity and subsequent ubiquitination of the client proteins. Therefore, the acetylation of HSP90 promotes FLT3 ubiquitination and enables proteasomal degradation of FLT3, which leads to inhibition of FLT3 downstream signaling. Preclinical studies show that IHCH9033 maintains efficacy in both leukemic cell lines and patient samples that are resistant to FLT3 inhibitors. When combined with an FLT3 inhibitor such as quizartinib, IHCH9033 produces a synergistic anti-leukemic effect, resulting in tumor regression in FLT3-ITD/TKD AML xenograft models and patient-derived xenografts [38].

Moreover, compound 22d, a selective histone deacetylase 8 (HDAC8) inhibitor, has been shown to decrease resistance of FLT3-mutated AML cells to quizartinib [37]. This agent was observed to reduce engraftment of primary FLT3-ITD mutant AML cells in mice treated with FLT3 inhibitors such as quizartinib. The combination of 22d with FLT3 inhibitors may therefore represent a promising therapeutic approach to overcoming resistance in FLT3-ITD AML.

Recently, dual HDAC/FLT3 inhibitors have been developed using a rational structure-based drug design strategy. Compound 25h is a dual FLT3 and HDAC inhibitor with potent anti-proliferative effects against both FLT3-mutated leukemic cells and cells resistant to FLT3 inhibitors [39]. Preclinical findings indicate that compound 25h demonstrates increased anti-proliferative activity against MOLM-13 cells, which are resistant to FLT3 inhibitors, in comparison to gilteritinib, vorinostat and their combination [39]. This compound achieves a more robust anti-tumor effect by targeting the phosphorylated ERK pathway than either single agents or their combinations.

### 7.4. CDK4/6 Inhibitors

CDK4/6 inhibitors specifically target cyclin-dependent kinases 4 and 6, which are crucial for cell cycle progression. Although these agents are mainly utilized in breast cancer, their potential is being explored in AML, particularly in combination with other therapies (Table 1 and Figure 1) [40]. 

FLT3 inhibitors may indirectly influence the CDK4/6 pathway; in contrast, CDK4/6 inhibitors directly inhibit the pathway and suppress cell differentiation [41]. Certain FLT3 inhibitors such as sorafenib and quizartinib may lead to resistance via distinct FLT3 TKD mutations like D835 and F691. Consequently, combining FLT3 inhibitors with CDK4/6 inhibitors has proven beneficial for overcoming FLT3 inhibitor resistance [49].

Palbociclib, a CDK4/6 inhibitor, has shown potential efficacy in FLT3-mutated AML treatment. It is able to induce cell cycle arrest and trigger apoptosis in FLT3 inhibitor-resistant cells. Preclinical studies with FLT3–D835Y-positive leukemic cells have demonstrated that palbociclib reduces cell viability and inhibits colony formation in methylcellulose, accompanied by a significant decrease in AKT and AURK mRNA levels, thus providing a strategy to overcome FLT3 inhibitor resistance [50].

Abemaciclib also acts by inhibiting CDK4/6 proteins, thereby affecting cell cycle progression during the G1 phase [51,52]. Recent in vitro evidence indicates that abemaciclib in combination with gilteritinib can help reverse FLT3 inhibitor resistance [52].

Additionally, AMG925 functions as a dual inhibitor targeting both CDK4/6 and FLT3. In a recent investigation of AMG925’s anti-leukemic effects, it suppressed cell growth and triggered apoptosis in both FLT3-mutant and FLT3 WT leukemic cell lines [53]. The compound demonstrated superior activity in cells harboring ITD, D835Y and ITD/D835Y double mutations. Mechanistically, AMG925 impaired CDK4-dependent phosphorylation of retinoblastoma protein, inhibited FLT3 activation and suppressed both FLT3-dependent and -independent signaling through the AKT/mTOR, MEK/ERK and STAT5 pathways [54]. These findings suggest that simultaneous inhibition of CDK4/6 and FLT3 may help overcome resistance mechanisms associated with FLT3 inhibitors in AML.

### 7.5. Dual JAK2 and FLT3 Inhibitors

Activation of JAK2 mutations, such as V617F, results in phosphorylation and activation of STAT5 independent of FLT3 activation [55]. For example, the JAK2 V658F mutation can stimulate the CSF2RB pathway, circumventing FLT3 signaling and promoting resistance to FLT3 inhibitors (Table 1 and Figure 1).

Recently, dual JAK2 and FLT3 inhibition has emerged as a promising therapeutic approach for AML patients with FLT3 mutations. The rationale is based on findings that FLT3 mutations can enhance JAK2 signaling, while JAK2 inhibitors can mitigate FLT3 inhibitor resistance [42]. Furthermore, this treatment approach allows for the simultaneous targeting of FLT3, JAK2 and other resistance-associated pathways. Therefore, dual inhibition may be advantageous in achieving prolonged responses in FLT3-mutated AML. 

Pacritinib is a novel dual JAK2/FLT3 inhibitor exhibiting equipotent activity against both FLT3 and JAK2. The agent inhibits FLT3 phosphorylation and downstream STAT5, MAPK and PI3K signaling pathways in both FLT3-ITD and WT-FLT3 cells, which results in overcoming resistance to FLT3 inhibition [56]. In a previous preclinical study, pacritinib was found to inhibit JAK2-mediated cellular signaling through activated JAK2 pathways [57]. Additionally, Hart et al. demonstrated that pacritinib decreased FLT3 phosphorylation and downstream STAT5 signaling in leukemic cells, thereby suppressing tumor growth in MOLM-13 cells [58].

The JAK2 inhibitor momelotinib exhibited equipotent activity comparable to FLT3 inhibition. It demonstrated potent FLT3 inhibitory effects on FLT3-ITD leukemic cells and effectively suppressed FLT3 inhibitor-resistant cells harboring A-loop mutations, such as D835, D839 and Y842 [59]. Furthermore, it was able to inhibit resistant cells driven by growth factor and hematopoietic cytokine-activated JAK2 signaling.

Pluripotin showed significant inhibitory activity against FLT3, BCR-ABL, JAK2, Ras-GAP and ERK [60]. This agent exhibited strong inhibitory effects in both mouse and human cells expressing FLT3-ITD, including those with secondary mutations such as the gatekeeper F691L. Furthermore, it was capable of suppressing adaptive resistance mediated by RAS-MAPK pathway activation, BCR-ABL and JAK2 signaling.

Compound 11r exhibits dual inhibitory activity toward JAK2 and FLT3, providing a therapeutic advantage in overcoming resistance associated with FLT3 inhibitors. A preclinical study demonstrated that it possesses potent anti-proliferative activity in FLT3-mutated MV4-11 cells [42]. 

### 7.6. PIM Inhibitor

PIM kinases, such as PIM-1, may contribute to the development of resistance to FLT3 inhibitors in AML with FLT3-ITD mutations. PIM-1 is upregulated downstream of FLT3-ITD and enhances cell proliferation and survival [43]. Upregulation of PIM-1 decreases the sensitivity to FLT3 inhibitors, diminishes their apoptotic effects and ultimately leads to resistance in FLT3-ITD AML cells. Dual inhibition of both FLT3 and PIM kinases with specific inhibitors may represent a more effective therapeutic approach (Table 1 and Figure 1). 

The pan-PIM inhibitor SGI-1776 can induce apoptosis in AML cells in a concentration-dependent manner. The activity of SGI-1776 corresponds with the inhibition of global RNA and protein synthesis, along with a reduction in MCL-1 transcript levels [61]. As a result, activation of SGI-1776 leads to Mcl-1 protein depletion and exerts cytotoxic effects in AML primary cells regardless of FLT3 mutation status, ultimately causing a decline in Mcl-1 protein levels. 

Concurrent administration of the pan-PIM inhibitor AZD1208 with FLT3 inhibitors suppressed in vitro proliferation of FLT3-ITD cell lines, but had no effect on WT-FLT3, through downregulation of Mcl-1 [62,63]. Using a combination of AZD1208 and the FLT3 inhibitor quizartinib, researchers observed reduced proliferation of MV4-11 cells carrying FLT3-ITD, improved survival outcomes, increased apoptosis in FLT3-ITD primary leukemic blasts and diminished colony formation of FLT3-ITD-positive AML cells [64].

### 7.7. CXCR4 Inhibitors

The CXCL12/CXCR4 axis plays a central role in cell migration, adhesion and survival within the BM microenvironment. Leukemic cells that evade therapy by utilizing survival signals from CXCL12/CXCR4 pathways are at increased risk for relapse [44]. Notably, CXCL12/CXCR4 signaling can activate pathways such as MAPK and ERK, which can diminish the efficacy of FLT3 inhibitors (Table 1 and Figure 1). 

LY2510924 antagonized CXCR4 on the cell surface without inducing apoptosis. This agent effectively reversed stroma-mediated resistance to quizartinib primarily by targeting the MAPK pathway [65]. In murine models of established FLT3-ITD-AML, LY2510924 facilitated sustained mobilization and differentiation of leukemia cells, enhancing the efficacy of quizartinib, while eliciting only transient effects on normal hematopoietic cells in immune-competent mice [66].

GMI-1359, a dual CXCR4/E-selectin antagonist, has exhibited potential in overcoming resistance to FLT3 inhibitors in AML [67]. GMI-1359 addresses resistance to FLT3 inhibitors by simultaneously targeting E-selectin and CXCR4, two adhesion molecules that FLT3 inhibitors may elevate, facilitating the retention of leukemia stem cells in the BM niche [68]. By blocking these signaling pathways, GMI-1359 disrupts the protective BM environment, thereby intensifying the anti-leukemic efficacy of FLT3 inhibitors such as quizartinib. 

### 7.8. BET Inhibitors

BET inhibitors reduce expression of pro-survival genes, including MYC and BCL2, though they have thus far shown only modest single-agent clinical efficacy (Table 1 and Figure 1) [69]. Recent data indicate that the novel 4-azaindole derivative PLX51107 exhibits both in vitro and in vivo BET-inhibitory activity. In experimental models, co-administration of BET and FLT3 inhibitors generated synergistic anti-leukemic effects in murine xenograft models of FLT3-ITD AML, as well as in primary FLT3-ITD AML cell specimens. The observed synergism was closely correlated with reduced transcription of survival genes like MYC.

Lee et al. demonstrated that the novel 4-azaindole derivative PLX51107 possesses BET-inhibitory properties both in vitro and in vivo. Mice receiving the quizartinib-PLX51107 regimen exhibited significantly greater inhibition of leukemic cell proliferation compared to those treated with quizartinib alone [45]. The agent’s short plasma half-life permitted intermittent target inhibition, improving tolerability while counteracting the protective influence of the BM microenvironment.

JQ1 demonstrated marked cytotoxic effects against BaF3 cells expressing FLT3-ITD alone or in combination with FLT3-TKD mutations, such as F691L or D835V [70]. Although FLT3-ITD-F691L and FLT3-ITD-D835V mutations confer resistance to FLT3 inhibitors like quizartinib and ponatinib, these mutations result in cellular dependence on the BET protein-regulated transcriptome for proliferation and survival.

According to the data, JQ1, a BET inhibitor, was able to overcome resistance to FLT3 inhibitors and synergistically potentiate anti-leukemic activity. In particular, the combination of JQ1 with FLT3 inhibitors enabled targeting of FLT3-ITD mutations, helping to overcome resistance to FLT3 inhibitors [70]. Furthermore, JQ1 exerted its effects by targeting alternative signaling pathways essential for leukemic cell survival, thereby circumventing FLT3 inhibitor resistance.

PLX51107 is a BET inhibitor that demonstrated synergistic effects when used in combination with FLT3 inhibitors in preclinical FLT3-mutated AML models [70]. Although this agent did not directly inhibit FLT3, it targeted MYC protein, a critical downstream effector of FLT3 signaling. A recent preclinical study showed that PLX51107 could bypass resistance mechanisms by inhibiting MYC protein [70]. Additionally, the combination of PLX51107 with FLT3 inhibitors resulted in enhanced cytotoxicity against FLT3-ITD AML cells. This therapeutic strategy may offer a practical approach to overcoming microenvironment-mediated resistance to FLT3 inhibition [70].

### 7.9. Multi-Kinase Inhibitors

Pexidartinib has been characterized as a triple-kinase inhibitor targeting KIT, colony-stimulating factor-1 receptor and FLT3-ITD in certain malignancies [71]. This agent was evaluated for its capacity to overcome resistance to other FLT3 inhibitors, particularly in the presence of the gatekeeper mutation F691L [72]. However, the anti-leukemic efficacy of pexidartinib was substantially affected by other point mutations, especially those within the A-loop, such as D835. Consequently, its effectiveness was reduced in cases with additional residue mutations, especially those occurring in the activation loop such as D835.

Dasatinib, a multi-targeted tyrosine kinase inhibitor, can counteract some forms of resistance to FLT3 inhibitors in AML cells harboring the FLT3-ITD mutation [73]. Notably, dasatinib mitigates resistance to FLT3 inhibitors like quizartinib, which emerges due to AML cell interactions with the BM microenvironment [74]. Current data indicate that dasatinib effectively overcomes BM microenvironment-mediated resistance even though it does not directly inhibit the FLT3 mutation. This resistance is suppressed through inhibition of STAT5 activation and suppression of glycolysis, which ultimately increases the sensitivity of cells to quizartinib.

Foretinib (GSK1363089) is a multi-kinase inhibitor that targets several receptor tyrosine kinases, including c-Met, vascular endothelial growth factor receptor-2, KIT, FLT3 and platelet-derived growth factor receptor-β [75]. Foretinib has been demonstrated to bind directly to FLT3, resulting in strong inhibition of FLT3 signaling. Consequently, this agent can suppress proliferation and induce apoptosis in human AML cell lines and primary AML cells harboring FLT3-ITD mutations [76]. Additionally, foretinib exhibited substantial efficacy against secondary FLT3-ITD mutations arising after prior quizartinib and gilteritinib exposure.

Dubermatinib (TP-0903) was developed to inhibit the Axl receptor tyrosine kinase, as well as FLT3, ABL, JAK2 and aurora A/B proteins. In a recent investigation, this agent demonstrated notable activity against protein kinases involved in STAT, AKT and ERK signaling [76]. The data indicated that dubermatinib is effective in a range of drug-resistant FLT3 mutant AML models, including those with the FLT3 F691L mutation and those influenced by BM microenvironment-mediated factors. Furthermore, dubermatinib exhibited preclinical activity in AML models carrying FLT3-ITD combined with isocitrate dehydrogenase 2 and NRAS co-mutations. Thus, dubermatinib displayed potent multi-kinase inhibitory activity against various drug-resistant AML models.

Luxeptinib (CG-806) demonstrated inhibitory effects on FLT3/BTK, suppressed autophagy and induced AML cell death by co-targeting FLT3/BTK/aurora kinases [77]. Recent studies have shown that luxeptinib has significant anti-leukemic effects in AML irrespective of FLT3 mutation status [78]. The compound markedly suppressed leukemic cell proliferation in an in vivo patient-derived xenograft murine model of FLT3 inhibitor-resistant FLT3-ITD/TKD double-mutant primary AML.

CHMFL-FLT3-122 showed selective inhibition of FLT3 kinase over BTK [79]. This agent significantly reduced the proliferation of FLT3-ITD-positive MV4-11 leukemic cell lines [80]. Moreover, a distinct inhibitory action was observed between FLT3 kinase and c-KIT kinase in TEL-fusion isogenic BaF3 cells. Accordingly, CHMFL-FLT3-122 robustly targeted FLT3-ITD-mediated signaling, leading to cell cycle arrest in the G0/G1 phase and induction of apoptosis.

GNF-7, initially reported as a BCR-ABL1 inhibitor, demonstrates a distinct inhibitory effect on FLT3 kinase and effectively suppresses FLT3 phosphorylation and associated downstream signaling pathways in FLT3-ITD cells [81]. Thus, GNF-7 represents a novel therapeutic approach as a multitargeted kinase inhibitor for the treatment of AML. Recent studies evaluated the survival benefits of GNF-7 in vivo using mouse models of Ba/F3 cells transformed with FLT3-ITD and FLT3-ITD/F691L mutation [82]. GNF-7 suppressed the proliferation of Ba/F3 cells expressing FLT3-ITD and showed robust anti-leukemic activity against FLT3-ITD AML samples. Furthermore, the compound was shown to interact with FLT3 protein and inhibit key downstream signaling pathways including STAT5, PI3K/AKT and MAPK/ERK. Notably, GNF-7 displayed significant inhibitory effects against FLT3-ITD/F691L, a mutation conferring resistance to quizartinib or gilteritinib. Critically, GNF-7 demonstrated a pronounced cytotoxic effect on leukemic stem cells, markedly prolonged survival in the PDX model and exhibited therapeutic efficacy comparable to gilteritinib.

KX2-391, functioning as a dual FLT3 and tubulin inhibitor, has been evaluated for its efficacy and underlying mechanisms in overcoming drug-resistant FLT3-ITD-TKD mutations in AML [83]. It demonstrated strong growth inhibition and induced apoptosis in AML cell lines with FLT3-ITD mutations and quizartinib-resistant mutations at the D835 and F691 sites [83]. Consequently, KX2-391 significantly extended the survival of leukemic cells with the FLT3-ITD-F691L mutation. It also markedly suppressed the proliferation of AML cells harboring FLT3-ITD-D835Y.

CCT241736 is a dual inhibitor targeting FLT3 and aurora kinases, and it has demonstrated potential in counteracting resistance to other FLT3 inhibitors in AML [84]. It holds promise as a therapeutic option for patients with FLT3-ITD and FLT3-TKD mutations who have acquired resistance to FLT3 inhibitors such as sorafenib and quizartinib. The compound exhibits substantial anti-FLT3 and aurora kinase activity and selectivity, supporting its development as a candidate in clinical trials for FLT3-ITD and TKDmut AML patients unresponsive to prior therapies.

Wu-5 selectively decreased the viability of both FLT3 inhibitor-sensitive and -resistant FLT3-ITD-positive AML cells. In earlier research, this agent induced apoptosis in MV4-11, Molm13 and MV4-11R cells in a dose-dependent manner by promoting proteasome-mediated degradation of FLT3-ITD and suppressing compound C AMPKα [85]. Moreover, combined administration of Wu-5 and crenolanib resulted in synergistic cell death in FLT3-ITD-positive cells by reducing levels of both FLT3 and AMPKα proteins.

Mer tyrosine kinase (MERTK) receptor is recognized to be overexpressed in most AML cases and plays a role in the development of leukemia [86]. MRX-2843 is a type 1 small-molecule multi-kinase inhibitor that blocks the activation of both MERTK and FLT3, as well as their downstream pathways [87]. A preclinical study demonstrated that MRX-2843 retained efficacy against quizartinib-resistant FLT3-ITD-mutant proteins with clinically relevant alterations at D835 or F691 in xenograft models of quizartinib-resistant AML [87].

## 8. Conclusions

Although FLT3 inhibitors have significantly influenced the management of AML with FLT-ITD mutations, relapse or refractory disease remains a leading cause of therapeutic failure.

Recently, several mechanisms of resistance to FLT3 inhibition have been identified and appear to be heterogeneous among individual AML patients. Improved understanding of resistance mechanisms to FLT3 inhibition and associated cellular pathways could aid in the development of individualized strategies to overcome resistance.

This suggests that specific, targeted strategies for several actionable resistance mechanisms are necessary in AML patients. In addition, the emergence of alternative signaling pathways necessitates the development of novel FLT3 agents or drugs effective against these bypass mechanisms. Consequently, recently developed FLT3-associated TKIs could be useful for controlling resistant leukemic clones through their predicted mechanisms of resistance.

The advancement of diagnostic technologies for identifying resistance mechanisms in AML, along with ongoing clinical trials to validate new agents that overcome resistance to FLT3 inhibitors, may lead to substantial progress in individualized therapeutic strategies for patients with resistant leukemic clones.

## Figures and Tables

**Figure 1 cells-14-01526-f001:**
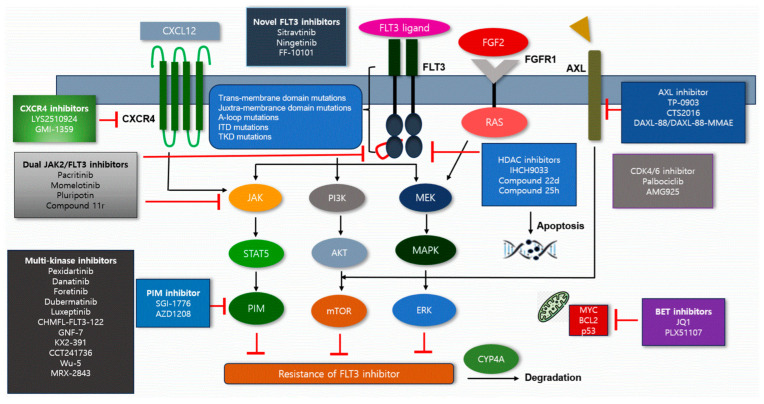
Overcoming therapeutic agents for resistance of FLT3 inhibitors in AML.

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
