# Peer review of "Counter-Therapeutic Strategies for Resistance of FLT3 Inhibitors in Acute Myeloid Leukemia"

_cells, 2025, doi:10.3390/cells14191526_

Round 1

Reviewer 1 Report

Comments and Suggestions for Authors

This review article focused on FLT3 mutations in acute myeloid leukemia (AML).

Acute myeloid leukemia (AML) comprises a heterogeneous group of aggressive blood cell cancers that arise from clonal expansion of malignant hematopoietic precursor cells in the bone marrow. The leukemic cells interfere with production of normal blood cells, causing anemia, infection, bleeding, and other symptoms and complications.

According to European LeukemiaNet (ELN) guidelines, AML should be diagnosed and classified according to criteria defined by either the International Consensus Classification (ICC) or the World Health Organization 5th edition (WHO5). Both the ICC and WHO5 models are based on cytogenetic and molecular features of AML blasts, and they supersede earlier classification schemes.

In AML, the prognosis depends of several factors (favorable, intermediate, adverse).

Regarding FLT3 and prognosis:

Favorable: Mutated NPM1 without FLT3-ITD

Intermediate: a) Mutated NPM1 with FLT3-ITD; b) Wild-type NPM1 with FLT3-ITD (without an adverse-risk genetic lesion)

Regarding the special scenarios of AML with mutated FLT3: For patients with FLT3-mutated AML, we recommend adding either midostaurin (for any FLT3 mutation) or quizartinib (for FLT3 with internal tandem duplication [ITD]) to 7+3 therapy. Adding a FLT3 inhibitor to 7+3 therapy achieves superior survival with little incremental toxicity in patients with mutated FLT3.

Of note, the US Food and Drug Administration (FDA) and the European Medicines Agency (EMA) approved midostaurin in combination with chemotherapy for adults with newly diagnosed AML with mutated FLT3. The FDA approved quizartinib in combination with 7+3 therapy for AML that is positive for FLT3-ITD but not for other FLT3 mutations; quizartinib is available only through a Risk Evaluation and Mitigation Strategy (REMS) program in the United States.

Main comment: This article thoroghly described many drugs that target FLT3 in AML. However, the review focused in many other makers that are related to FLT3 pathway. Since the title is "Counter therapeutic strategies for resistance of FLT3 inhibitors in acute myeloid leukemia", this may mislead the potential readers. I would recommend to highlight the word "pathway" and/or describe that it is a more comprehensive review.

Additional comments:

(1) Please describe AML with more details. In the current text, no information is provided. If the article focuses on readers who are expert of AML, then it is fine, but a description of AML may broaden the audience.

For example:

Epidemiology
Pathogenesis
Clinical presentation
Diagnosis and classficiation
Prognosis
FIT, younger and "less-FIT" or older patients
Monitoring
Relapsed/Refractory disease
Clinical trials

(2) As I understand, Figure 1 described the FLT3 pathway and the drug targets. Is this the complete FLT3 pathway? 

Please look into this review that shows many useful tables and figures:

Front. Oncol., 23 December 2020. Sec. Hematologic Malignancies. Volume 10 - 2020 | https://doi.org/10.3389/fonc.2020.612880

(3) Regarding Table 1. Please confirm if it is necessary to add references within the Table.

(4) Regarding the drugs that are being described. Which one have FDA approval? Which ones are just tested on cell lines or in vivo models or in humans?

(5) Are FLT3 mutations both of gain of function and others of loss of function? Distinction is relevant as downstream effect change and additional requirement of drug changes.

(6) Please describe the general mutational profile of AML, and drug targeting.

(7) Please show FLT3 protein structure and location of the mutations. 3D model or 2D model.

Author Response

(Reviewer 1)

Main comment: This article thoroghly described many drugs that target FLT3 in AML. However, the review focused in many other makers that are related to FLT3 pathway. Since the title is "Counter therapeutic strategies for resistance of FLT3 inhibitors in acute myeloid leukemia", this may mislead the potential readers. I would recommend to highlight the word "pathway" and/or describe that it is a more comprehensive review.

(Answer) We has already decided to retain the current title. please understand me.

Additional comments:

(1) Please describe AML with more details. In the current text, no information is provided. If the article focuses on readers who are expert of AML, then it is fine, but a description of AML may broaden the audience.

For example:

Epidemiology

Pathogenesis

Clinical presentation

Diagnosis and classficiation

Prognosis

FIT, younger and "less-FIT" or older patients

Monitoring

Relapsed/Refractory disease

Clinical trials

(Answer) Our suject is “overcoming strategy for resistance of FLT3 inhibitors.”

We focus on the resistance of FLT3 inhibitors. please understand the subject.

(2) As I understand, Figure 1 described the FLT3 pathway and the drug targets. Is this the complete FLT3 pathway?

Please look into this review that shows many useful tables and figures:

Front. Oncol., 23 December 2020. Sec. Hematologic Malignancies. Volume 10 - 2020 | https://doi.org/10.3389/fonc.2020.612880

(Answer) I think that we could focus on our subject. The above mentioned article is out of tune with our subject.

(3) Regarding Table 1. Please confirm if it is necessary to add references within the Table.

(Answer) Okay, I will add the references.

(4) Regarding the drugs that are being described. Which one have FDA approval? Which ones are just tested on cell lines or in vivo models or in humans?

(Answer) In our article, we described the development of molecular diagnostic method to confirm primary and secondary resistance, and the appropriate treatments. would Let me know for the confirmed diagnostic method and appropriate FDA approved drugs. I know the agenets stay on a laboratory level for the most part.

(5) Are FLT3 mutations both of gain of function and others of loss of function? Distinction is relevant as downstream effect change and additional requirement of drug changes.

(Answer) The commend is very difficult to talk about the suject in article.

Can you explain what the topic mentioned above has to do with our paper?

(6) Please describe the general mutational profile of AML, and drug targeting.

(Answer) This article is focusing in overcoming strategy to resistance of FLT3 inhibitors. generation mutation is not focus, and far from subjet of our review point.

(7) Please show FLT3 protein structure and location of the mutations. 3D model or 2D model.

(Answer) Tell me why you need a FLT3 2D or 3D model. This article is not focusing to FLT3 mutations. Overcoming strategy is focus.

Reviewer 2 Report

Comments and Suggestions for Authors

This manuscript reviews the relevance of the FLT3 receptor in the growth of acute myeloid leukemia and the therapeutic means to overcome FLT3-inhibitor resistance.

The authors concisely describe the role of FLT3 in AML, the alteration of this tyrosine kinase receptor, and the mechanisms of resistance to FLT3-inhibitor treatment in AML.

This review is well-written and clear. 

Minor revision is needed to help the reader get the message of this manuscript.

Figure 1 is a summary of this review, which is very dense with information.  It is placed at the beginning of this review, and this leads to the need to go back through the text to look at the point of action of a given inhibitor. Please cut into two or more pieces, inserting a brief legend (the present legend is the title of this figure). 

The authors made a comprehensive list of the old and new FLT3 inhibitors in Table 1. To this table should be added the references dealing with the mechanism of action and possibly the clinical trials (if any) in which these inhibitors have been applied.

Some notes on the use of FLT3 inhibitors in association with chemotherapy should be considered. Similar to the possible effects of FLT3 inhibitors on the immune system.

Author Response

(Reviewer 2)

Minor revision is needed to help the reader get the message of this manuscript.

Figure 1 is a summary of this review, which is very dense with information. It is placed at the beginning of this review, and this leads to the need to go back through the text to look at the point of action of a given inhibitor. Please cut into two or more pieces, inserting a brief legend (the present legend is the title of this figure).

(Answer) Editor could change to the field.

The authors made a comprehensive list of the old and new FLT3 inhibitors in Table 1. To this table should be added the references dealing with the mechanism of action and possibly the clinical trials (if any) in which these inhibitors have been applied.

Some notes on the use of FLT3 inhibitors in association with chemotherapy should be considered. Similar to the possible effects of FLT3 inhibitors on the immune system.

(Answer) Thank you. We will add the references.

Reviewer 3 Report

Comments and Suggestions for Authors

This is a comprehensive and well written review. The topic  is relevant in a type of acute myeloid leukemia. The only drawback would be the narrowness of the topic. The table 1 is informative. However in the Introduction it would be helpful a figure showing an scheme of FLT3 with the domains where the most common mutations map (ITD. TKD, A-loop).  Also, it is needed a clear statement on the frequencies of mutation in these domains in AML

I have other minor comments

  • Line 40-41 Reference needed to support this sentence
  • Lines 101-3: BAD is not showed in the Figure 1
  • Line 151: "RAS-driven oncogenic mutations..."Which mutations are involved here? Frequency o RAS mutation ?
  • Line 243: Why the over-acetylation FTL3 drives its ubiquitination This point needs to de developed or explained

Author Response

(Reviewer 3)

I have other minor comments

Line 40-41 Reference needed to support this sentence

(Answer) we added reference 4.

Lines 101-3: BAD is not showed in the Figure 1

(Answer) please understand the information of figure. In addition to BAD, other pathways are also difficult to descrbie due to the narrowed figure area.

Line 151: "RAS-driven oncogenic mutations..."Which mutations are involved here? Frequency o RAS mutation ?

(Asnwer) RAS-driven oncogenic mutations change to RAS/MAPK pahways

However, The frequnecy is not important in the sentence. please understand

Line 243: Why the over-acetylation FTL3 drives its ubiquitination This point needs to de developed or explained

(Answer) we added the information in line 241-243. Than you